# How Sampling Impacts the Robustness of Stochastic Neural Networks

**Sina Däubener and Asja Fischer**
Department of Computer Science
Ruhr University Bochum, Germany
{sina.daeubener, asja.fischer}@rub.de

## Abstract

Stochastic neural networks (SNNs) are random functions whose predictions are gained by averaging over multiple realizations. Consequently, a gradient-based adversarial example is calculated based on one set of samples and its classification on another set. In this paper, we derive a sufficient condition for such a stochastic prediction to be robust against a given sample-based attack. This allows us to identify the factors that lead to an increased robustness of SNNs and gives theoretical explanations for: (i) the well known observation, that increasing the amount of samples drawn for the estimation of adversarial examples increases the attack's strength, (ii) why increasing the number of samples during an attack can not fully reduce the effect of stochasticity, (iii) why the sample size during inference does not influence the robustness, and (iv) why a higher gradient variance and a shorter expected value of the gradient relates to a higher robustness. Our theoretical findings give a unified view on the mechanisms underlying previously proposed approaches for increasing attack strengths or model robustness and are verified by an extensive empirical analysis.

## 1   Introduction

Since the discovery of adversarial examples [Biggio et al., 2013, Szegedy et al., 2014], a significant amount of research was dedicated to hinder attacks [e.g. Madry et al., 2018, Papernot et al., 2016, Zhang et al., 2019], to enhance attack strategies [e.g. Athalye et al., 2018, Akhtar and Mian, 2018, Carlini and Wagner, 2017, Uesato et al., 2018] or to derive ways to certify model robustness [e.g. Cohen et al., 2019, Lécuyer et al., 2019]. Robustness guarantees often specify an $\epsilon$-ball around input points in which perturbations do not lead to a label change [Hein and Andriushchenko, 2017, Croce and Hein, 2020]. The maximal possible radius of such an $\epsilon$-ball corresponds to the distance of the input point to the nearest decision boundary, which on the other hand is equal to the length of the smallest perturbation vector that leads to a misclassification (c.f. figure 1a)). Such a robustness analysis assumes that the decision boundaries are fixed and that the attacker is able to estimate (at least approximately) this minimal perturbation vector, which is a reasonable assumption for deterministic networks but usually does not hold for stochastic neural networks (SNNs).

Stochastic neural networks, and stochastic classifiers more generally, are random functions and predictions are given by the expected value of the random function for the given input. In practice, this expectation is usually not tractable and hence it is approximated by averaging over multiple realizations of the random function. This approximation leads to the challenging setting where predictions, decision boundaries, and gradients become random variables themselves. Hence, under an adversarial attack, the decision boundaries used for calculating the adversarial example and those used when predicting the label of the resulting adversarial example differ. This means that the attacker can not estimate the optimal perturbation direction i.e., the direction to the closest decision boundary of the network that will be sampled during inference c.f. figure 1 b), c).

36th Conference on Neural Information Processing Systems (NeurIPS 2022).

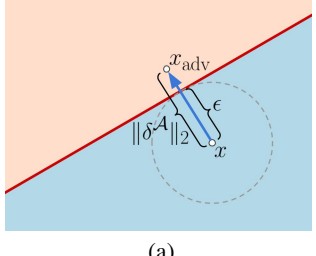 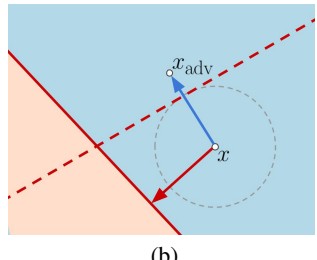 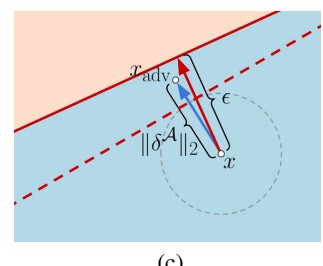

|     |     |     |
| --- | --- | --- |
| (a) | (b) | (c) |

Figure 1: (Un-)successful attacks on a binary stochastic classifier with a linear decision boundary. a) an adversarial example $x_{\text{adv}} = x + \delta^{\mathcal{A}}$ is created by shifting $x$ in the direction of the closest decision boundary, indicated by the blue arrow. $\epsilon$ is the minimal needed attack length and the attack can only be an adversarial example if $\|\delta^{\mathcal{A}}\|_2 \geq \epsilon$ holds. b) and c) show the stochastic decision boundaries during attack (dashed) and inference (solid). The red arrows indicate the shortest direction to the latter, respectively. In b) $\delta^{\mathcal{A}}$ moves $x$ even further away from the decision boundary used during inference, while in c) the magnitude of $\delta^{\mathcal{A}}$ is too short to result in a successful attack.

In this paper, we study how robustness of SNNs arises from this misalignment of the attack direction and the optimal perturbation direction during inference that results from the stochasticity inherent to stochastic classifiers.

We make the following contributions: First, we derive a sufficient condition for a SNN prediction relying on one set of samples to be robust against an attack that was calculated on a second set of samples. Second, we discuss how model properties and sample sizes impact this condition which does not only allows us to answer the questions stated in the abstract but also to explain the success of recently proposed defense mechanism from a simple unifying geometric perspective. Lastly, we conduct an empirical analysis that demonstrates that the novel theoretical insights perfectly match what we observe in practice.

## 2 Related work

Several works proposed stochastic defense mechanism to increase adversarial robustness [e.g. Raff et al., 2019, Xie et al., 2018]. Athalye et al. [2018] linked their success to gradient obfuscating during the attack and showed that increasing the number of samples for approximating the gradient during attack leads to a sever decrease in adversarial accuracy.

However, SNNs were still found to have an increased robustness even w.r.t. stronger attacks [Yu et al., 2021, He et al., 2019, Eustratiadis et al., 2021, Jeddi et al., 2020]. Their success was attributed to different effects of stochasticity, e.g. model smoothing [Liu et al., 2018, Addepalli et al., 2021] or diversification of the gradients [Lee et al., 2022, Bender et al., 2020]. While a lot of work analyzed the robustness of deterministic neural networks [e.g. Madry et al., 2018, Croce et al., 2019, Croce and Hein, 2020, Dabouei et al., 2020, Yang et al., 2022], the robustness of SNNs is less well understood. One line of research focused on the robustness of Bayesian neural networks (BNNs). Wicker et al. [2020] certified robustness of BNNs using interval bound propagation techniques which they later also employed to derive guarantees for the robustness of BNNs with modified adversarial training [Wicker et al., 2021]. Moreover, Carbone et al. [2020] investigate robustness in the infinite-width infinite-sample limit. Lastly, Pinot et al. [2019] derived theoretical robustness guarantees for randomized networks, where the randomization is based on additive noise from an exponential family distribution. This leads to a generalization of the robustness guarantees derived previously by Lécuyer et al. [2019] and Cohen et al. [2019]. Those certified robustness guarantees specify the radius of an $\ell_2$-ball in which the prediction does not change. In contrast, our results specify the robustness that results from the difficulty to identify the ideal attack direction and imply that even for perturbations outside this confidence ball, a gradient-based attack has a chance of not being successful.

To the best of our knowledge, no existing theoretical analysis of SNNs explicitly discusses either the effect of stochasticity during inference nor the impact of the sample size during attack and prediction on the robustness.

## 3 Preliminaries

We first clarify the terminology before we state the main theoretical results of our paper.

**Stochastic classifiers** We use the term *stochastic classifiers* for all classifiers which have an inherent stochasticity through the use of random variables in the model. Formally, we define them as follows:

**Definition 3.1** (Stochastic classifiers). A stochastic classifier with $k$ classes corresponds to a function $f : \mathbb{R}^d \times \Omega^h \to \mathbb{R}^k$ that maps a pair $(x, \Theta)$ to the output $f(x, \Theta) = (f_1(x, \Theta), \dots f_k(x, \Theta))^T$, where $x \in \mathbb{R}^d$ is an input vector, $\Theta \in \Omega^h, \Theta \sim p(\Theta)$ is a random vector, and $f_c(x, \Theta)$ with $c \in \{1, \dots, k\}$ are the discriminant functions for each class. The prediction of a stochastic classifier for an input $x$ is given by $\mathbb{E}_{\Theta}[f(x, \Theta)]$ and the predicted class by $\arg\max_c \mathbb{E}_{\Theta}[f_c(x, \Theta)]$.

This generic definition of stochastic classifiers covers linear models with random weights, but also more complicated methods like BNNs [Neal, 1996], infinite mixtures [Däubener and Fischer, 2020], Monte Carlo dropout networks [Gal and Ghahramani, 2016], randomized smoothing as proposed by Lécuyer et al. [2019][1], and any other class of neural networks which use stochasticity at the input level [e.g. Raff et al., 2019] or within the network [e.g. He et al., 2019, Jeddi et al., 2020, Liu et al., 2018, Yu et al., 2021, Eustratiadis et al., 2021]. In cases where $\mathbb{E}_{\Theta}[f(x, \Theta)]$ is not tractable — which is in practice usually the case — the prediction of the stochastic classifier is approximated by its Monte Carlo (MC) estimate $f^{\mathcal{S}}(x) := \frac{1}{S} \sum_{s=1}^{S} f(x, \theta_s)$, where the samples in the sample set $\mathcal{S} = \{\theta_1, \dots, \theta_S\}$ are drawn from $p(\Theta)$.

**Adversarial attacks on stochastic classifiers** Informally speaking, adversarial examples are inputs that are modified such that the network predicts wrong classes even though the changes to the inputs are not perceptible for a human. More precisely, let $x$ be an input with corresponding true label $y \in \{1, \dots, k\}$ that is classified correctly by the multi-class classifier $f(\cdot)$, that is $\arg\max_c f_c(x) = y$. We consider the most common attack form which aims at misclassifying $x$ by allowing for some predefined maximum magnitude of perturbation. That is, the attacker targets the optimization problem

$$maximize \quad \mathcal{L}(f(x + \delta), y) \ , \quad s.t. \quad \|\delta\|_p \leq \eta \ , \tag{1}$$

where $\mathcal{L}(\cdot, \cdot)$ is the loss function, $\|\cdot\|_p$ with $p \geq 1$ is the $\ell_p$-norm, and $\eta$ is the *perturbation strength*, i.e. the maximal allowed magnitude of the attack (see figure 1a) for an illustration). Common choices of loss functions include the cross-entropy loss and the negative margin loss $\mathcal{L}_{\text{margin}}(f(x + \delta), y) = -(f_y(x + \delta) - \max_{c \neq y} f_c(x + \delta))$. In practice, targeting the optimization problem in eq. (1) usually involves estimating $\delta$ by performing some kind of gradient-based optimization on the loss function [e.g. Goodfellow et al., 2015, Madry et al., 2018]. If $f^{\mathcal{S}}(\cdot)$ is a stochastic classifier (as introduced in the previous paragraph) approximated by its MC estimate, the loss gradient with respect to the input is stochastic as well and given by

$$\nabla_x \mathcal{L}(f^{\mathcal{S}}(x), y) = \nabla_x \mathcal{L}\left(\frac{1}{S} \sum_{s=1}^{S} f(x, \theta_s), y\right) \ . \tag{2}$$

Note, that for linear $\mathcal{L}$, as for example the margin loss for a fixed class $c$, the loss gradient of the mean prediction is equivalent to the mean of the loss gradients for single sample predictions. However, in general (e.g. for the cross entropy loss) this is not the case.

## 4 Geometrical robustness analysis

After clarifying our understanding of stochastic classifiers and on how gradient-based adversarial attacks are conducted on them, we are now able to present a simple but general geometrical, adversarial robustness analysis for stochastic classifiers. It is motivated by the following observation: Each time a stochastic classifier is used for a prediction, another set of realizations from the random vectors are

---

[1]Note, that, in contrast to Lécuyer et al. [2019] the variant of randomized smoothing proposed by Cohen et al. [2019] does not define the prediction of the SNN to be given by the expectation over $\Theta$ but by $\arg\max_{c \in \mathcal{Y}} P(f(x + \epsilon) = c)$, where $\epsilon \sim \mathcal{N}(0, \sigma^2 I)$. This makes our results not directly applicable to their networks. However, they can probably be transferred to the decision boundaries and attacks corresponding to their form of generating predictions.

drawn, resulting in another set of discriminant functions and corresponding decision boundaries. To put it into other words, the classifier gets a random variable itself. As a consequence, calculating a gradient-based adversarial attack for an input $x$ is done with respect to a drawn set of realizations $\mathcal{A} = \{\theta_1^a, \theta_2^a, \ldots, \theta_{S^\mathcal{A}}^a\}$, and thus $\delta$ from eq. (1) is specific for a realization $f^\mathcal{A}(x)$ of the classifier, which we specify by writing $\delta^\mathcal{A}$. During inference, the resulting adversarial example $x_{\text{adv}} = x + \delta^\mathcal{A}$ is then fed to another random classifier $f^\mathcal{I}$, which is based on a different set of realizations from the random vector $\mathcal{I} = \{\theta_1^i, \theta_2^i, \ldots, \theta_{S^\mathcal{I}}^i\}$. From this perspective, the prediction model $f^\mathcal{I}$ is robust against the attack if the distance from $x$ to the decision boundary (given by $f_y^\mathcal{I}(x) - f_{c \neq y}^\mathcal{I}(x) = 0$) in the *direction of $\delta^\mathcal{A}$* is larger than the length of $\delta^\mathcal{A}$, as illustrated in figures 1b) and c).

**Robustness conditions for stochastic attacks** For a classifier with linear discriminant functions, we are able to turn the previously described observation into a theorem in which we derive a sufficient and necessary condition for the prediction model to be robust against a given attack.

**Theorem 4.1** (Sufficient and necessary robustness condition for linear classifiers). *Let $f : \mathbb{R}^d \times \Omega^h \to \mathbb{R}^k$ be a stochastic classifier with linear discriminant functions and $f^\mathcal{A}$ and $f^\mathcal{I}$ be two MC estimates of the classifier. Let $x \in \mathbb{R}^d$ be a data point with label $y \in \{1, \ldots, k\}$ and $\arg\max_c f_c^\mathcal{A}(x) = \arg\max_c f_c^\mathcal{I}(x) = y$, and let $x_{adv} = x + \delta^\mathcal{A}$ be an adversarial example computed for solving the minimization problem (1) for $f^\mathcal{A}$. It holds that $\arg\max_c f_c^\mathcal{I}(x + \delta^\mathcal{A}) = y$ if and only if*

$$\min_{c \neq y} \tilde{r}_c^\mathcal{I} > \|\delta^\mathcal{A}\|_2 \ , \ \text{with} \tag{3}$$

$$\tilde{r}_c^\mathcal{I} = \begin{cases} \infty \ , & \text{if} \ \cos(\alpha_c^{\mathcal{I}, \mathcal{A}}) = \frac{\langle -\nabla_x(f_y^\mathcal{I}(x) - f_c^\mathcal{I}(x)), \delta^\mathcal{A} \rangle}{\|\nabla_x(f_y^\mathcal{I}(x) - f_c^\mathcal{I}(x))\|_2 \cdot \|\delta^\mathcal{A}\|_2} \leq 0 \\ \frac{f_y^\mathcal{I}(x) - f_c^\mathcal{I}(x)}{\|\nabla_x(f_y^\mathcal{I}(x) - f_c^\mathcal{I}(x))\|_2 \cdot \cos(\alpha_c^{\mathcal{I}, \mathcal{A}})} \ , & \text{otherwise} \ , \end{cases}$$

*where $\alpha_c^{\mathcal{I}, \mathcal{A}}$ is the angle between $-\nabla_x(f_y^\mathcal{I}(x) - f_c^\mathcal{I}(x))$ and $\delta^\mathcal{A}$.*

The proof which is based on Taylor expansion is given in supplement A. The conditions for $\cos(\alpha_c^{\mathcal{I}, \mathcal{A}})$ have a nice geometrical interpretation: An angle of more than 90° (which corresponds to a negative cosine value) indicates that the gradient $-\nabla_x(f_y^\mathcal{I}(x) - f_c^\mathcal{I}(x))$ and the perturbation $\delta^\mathcal{A}$ point into "opposite" directions and thus even for infinitely long moves into the direction of $\delta^\mathcal{A}$, the predicted label for $x$ will not change to class $c \neq y$. For positive cosine values, $\tilde{r}_c^\mathcal{I}$ specifies the distance to the decision boundary in the attack direction. It looks similar to the minimal distance to the decision boundary in a deterministic setting which is given by $\frac{f_y(x) - f_c(x)}{\|\nabla_x(f_y(x) - f_c(x))\|_2}$ and which is recovered if $\cos(\alpha_c^{\mathcal{I}, \mathcal{A}}) = 1$. A cosine value of one however can only occur if the gradient and perturbation direction are identical i.e., if the margin loss is used for calculating the attack direction and if attack and inference model are identical. In practice, the latter is almost surely not the case due to the finite sample approximation, and thus $\cos(\alpha_c^{\mathcal{I}, \mathcal{A}}) < 1$. This illustrates the robustness advantage induces by stochasticity.

The derived conditions may locally hold for classifiers which can be reasonably well approximated by a first-order Taylor approximation. To derive further guarantees, we can relax the linearity assumption by assuming discriminant functions which are $L$-smooth as defined in the following.

**Definition 4.1** ($L$-smoothness [Yang et al., 2022]). A differentiable function $f : \mathbb{R}^d \to \mathbb{R}^k$ is $L$-smooth, if for any $x_1, x_2 \in \mathbb{R}^d$ and any output $c \in \{1, \ldots, k\}$:

$$\frac{\|\nabla_{x_1} f_c(x_1) - \nabla_{x_2} f_c(x_2)\|_2}{\|x_1 - x_2\|_2} \leq L \ .$$

For such smooth discriminant function we can derive a sufficient (but not necessary[2]) robustness condition specified by the following theorem, which is proven in supplement A.

**Theorem 4.2** (Sufficient condition for the robustness of a L-smooth stochastic classifier). *In the setting of Theorem 4.1, let $f^\mathcal{A}$ and $f^\mathcal{I}$ be L-smooth (instead of linear) discriminant functions. Then it*

---

[2]The necessity of this condition is not given since for non-linear decision boundaries it is possible that behind a region of a different class there exists another region of class $y$ that an attack ends in if $\delta^\mathcal{A}$ is long enough.

*holds that* $\arg\max_c f_c^{\mathcal{I}}(x + \delta^{\mathcal{A}}) = y$ *if* $\min_{c \neq y} r_c^{\mathcal{I}} > \|\delta^{\mathcal{A}}\|_2$ *with*

$$
r_c^{\mathcal{I}} = \begin{cases} \infty \ , & \text{if } \|\nabla_x(f_y^{\mathcal{I}}(x) - f_c^{\mathcal{I}}(x))\|_2 \cdot \cos(\alpha_c^{\mathcal{I},\mathcal{A}}) + \frac{L}{2} \cdot \|\delta^{\mathcal{A}}\|_2 \leq 0 \ , \\[2ex] \frac{f_y^{\mathcal{I}}(x) - f_c^{\mathcal{I}}(x)}{\|\nabla_x(f_y^{\mathcal{I}}(x) - f_c^{\mathcal{I}}(x))\|_2 \cdot \cos(\alpha_c^{\mathcal{I},\mathcal{A}}) + \frac{L}{2} \cdot \|\delta^{\mathcal{A}}\|_2} \ , & \text{otherwise} \end{cases}
$$

*with* $\cos(\alpha_c^{\mathcal{I},\mathcal{A}})$ *as in theorem 4.1.*

**Factors influencing the robustness of stochastic classifiers**   Since in practice neither the parameter set $\mathcal{A}$ nor $\mathcal{I}$ is fixed, the quantity better describing the practical robustness of a stochastic classifier with linear discriminant functions is

$$
\mathbb{P}(\min_{c \neq y} \tilde{r}_c^{\mathcal{I}} > \|\delta^{\mathcal{A}}\|_2) \ , \tag{4}
$$

where $\tilde{r}_c^{\mathcal{I}}$ and $\delta^{\mathcal{A}}$ are random variables. For $L$-smooth models, replacing $\tilde{r}_c^{\mathcal{I}}$ by $r_c^{\mathcal{I}}$ leads to a lower bound on the robustness. Deriving an analytic expression for this probability is a hard problem. However, based on theorems 4.1 and 4.2 it becomes clear that a larger $\min_{c \neq y} \tilde{r}_c^{\mathcal{I}}$ relates to increasing the probability in eq. (4) and thus to an increased robustness. We note that $\tilde{r}_c^{\mathcal{I}}$ with $c \neq y$ grows with i) larger prediction margins $f_y^{\mathcal{I}}(x) - f_c^{\mathcal{I}}(x)$, ii) smaller gradient norms $\|\nabla_x(f_y^{\mathcal{I}}(x) - f_c^{\mathcal{I}}(x))\|_2$, and iii) larger angles $\alpha_c^{\mathcal{I},\mathcal{A}}$. Larger prediction margins and smaller gradient norms were also found to positively impact the robustness of deterministic networks [e.g. Ross and Doshi-Velez, 2018]. In contrast, the dependency on the angle is unique to the stochastic setting. We therefore focus on the analysis of this factor in the following.

**Analyzing the expected angle**   The angle $\alpha_c^{\mathcal{I},\mathcal{A}}$ depends on both terms $-\nabla_x(f_y^{\mathcal{I}}(x) - f_c^{\mathcal{I}}(x))$ (for which we use the shorthand $-\nabla_x f_{y-c}^{\mathcal{I}}(x)$ in the following) and $\delta^{\mathcal{A}}$. For further analysis, we first rewrite the gradient as

$$
-\nabla_x f_{y-c}^{\mathcal{I}}(x) = \nabla_x \left( \frac{1}{S^{\mathcal{I}}} \sum_{s=1}^{S^{\mathcal{I}}} -f_{y-c}(x, \theta_s^i) \right) = \frac{1}{S^{\mathcal{I}}} \sum_{s=1}^{S^{\mathcal{I}}} -\nabla_x f_{y-c}(x, \theta_s^i) \ .
$$

Let $\mu = \mathbb{E}_\Theta[-\nabla_x f_{y-c}(x, \theta_s^i)]$ and $\Sigma$ be the covariance of $-\nabla_x f_{y-c}(x, \theta_s^i)$, then it follows from the central limit theorem that for sufficiently many samples $-\nabla_x f_{y-c}^{\mathcal{I}}(x) \sim \mathcal{N}\left(\mu, \frac{\Sigma}{S^{\mathcal{I}}}\right)$. For simplicity let us assume that the attack is based on the margin loss and only one iteration of gradient ascent. In this case, $\delta^{\mathcal{A}} = \frac{\eta}{\|\hat{\delta}\|_2} \cdot \hat{\delta}$, with $\hat{\delta} \sim \mathcal{N}\left(\mu, \frac{\Sigma}{S^{\mathcal{A}}}\right)$. Note, that we can neglect the scaling of the attack vector, since the angle only depends on $\hat{\delta}$. Therefore, estimating the distribution of $\alpha_c^{\mathcal{I},\mathcal{A}}$ corresponds to estimating the distribution of the angle between two independent multivariate Gaussian random vectors with the same mean and (potentially) differently scaled versions of the same covariance.[3] It is known for vectors from normalized standard multivariate Gaussian distributions that the mode of the distribution of the angle between two random vectors is equal to 90° and that with increased dimension the concentration around this mode gets tighter [Cai et al., 2013]. Unfortunately, deriving a closed form expression for the distribution or expectation in the general case is a challenging task and beyond the scope of this paper. However, we conjecture that the expectation of the angle increases proportionally with the variance and anti-proportionally with the norm of the mean. We illustrate the intuition behind this hypothesis in figure 2. To empirically verify the correctness of this hypothesis we conducted an experiment where we estimated the expected angle between two identically distributed 1,000-dimensional Gaussian random vectors for different choices of means and diagonal covariances. More precisely, we sampled the mean and variances uniformly from $[0, t]$, where $t$ increased from zero to ten with step size 0.2, and estimated the expected angle based on 10,000 vector pairs drawn from the resulting distributions. Results are shown in figure 2 on the right side. As hypothesized, the smaller the length of $\mu$ and the higher the average variances, the higher the expected angle.

---

[3]In the case of more complicated attacks the mean will differ as well but we expect the findings of these section to generalize also to this scenario.

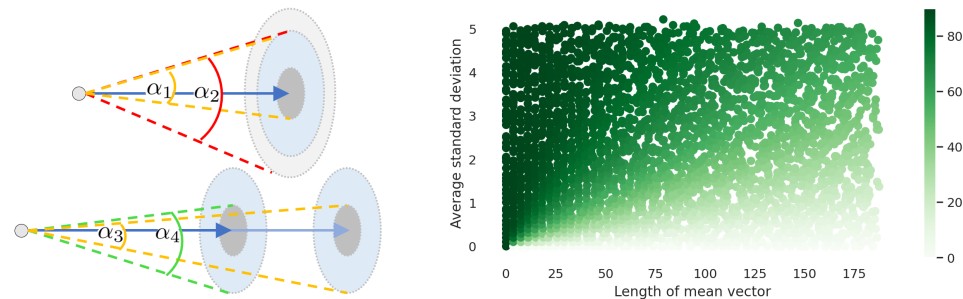

Figure 2: Dependence of the angle w.r.t mean and variance of the gradient. **Left, top**: Sketch of decreased variance which leads to smaller angles. The mean of the gradient is shown as the blue arrow. The circles indicate the areas of high probability for different scales of the covariance matrix. When the covariance is decreased (dark compared to light gray region) the maximum angle between vectors in these region to vectors from the blue region is decreased as well ($\alpha_1$ compared to $\alpha_2$). **Left, bottom**: Sketch of increased mean which leads to smaller maximal angles. When the covariance is fixed and the mean is increased (dark compared to light blue arrow) the maximum angle between vectors from high probability regions decreases ($\alpha_3$ compared to $\alpha_4$). **Right**: Expected angle between two identically distributed 1,000-dimensional Gaussian random vectors, for Gaussian with different means and variances. Expected angles were calculated based on 10,000 pairwise draws from the respective Gaussian distributions.

**Implications for the attack**   Based on the previous discussion, the only way the attacker can influence the probability of a successful attack, and in this sense the robustness of the model against the attack, is by increasing the amount of samples and thereby reducing the variance of the attack vector. A reduction of the variance leads to a decrease of the expected angle as described in the previous section. This gives a more elaborated explanation of what was often loosely described as finding the "correct" gradient direction in previous work [Athalye et al., 2018]. However, even if the attacker would be able to take infinitely many samples, and thus the variance of the attack direction would be reduced to zero, the expected value of the angle will be larger than zero because of the still existing stochasticity in the inference process. This stochasticity can even lead to an expected angle close to 90° if $\mu$ is short, and/or the covariance $\Sigma$ is high. That is, the advantage of obfuscating the optimal attack direction by incorporating stochasticity into the classifier can be decreased but not fully counterbalanced by taking more samples during the attack. This might explain the finding in He et al. [2019], that the accuracy under attack stagnates at a higher level than the deterministic counterpart when increasing the number of iterations of iterative gradient-based attack methods.

**Implications for the model**   From the perspective of the defender, the analysis of the angle shows that models with an increased gradient variance (which is often associated to a high prediction variance) and a small norm of the mean gradient are connected to larger values of $\alpha_c^{\mathcal{I},\mathcal{A}}$ and thus to a higher probability of unsuccessfully attacks. This explains why including the norm of the mean gradient, the gradient variance, or the angle between gradients as regularization terms in the training of SNNs, as for example proposed by Bender et al. [2020] and Lee et al. [2022][4], lead to an increased empirical robustness.

It would be naturally to suspect that increasing the number of samples used during inference also decreases the robustness, since it decreases the expected angle. However, this is not the case, as it is counterbalanced by an decrease of the norm of the gradient estimate (i.e. the second term in the denominator of $\tilde{r}_c$) with growing sample size.[5] This can be seen by rewriting the expected denominator with respect to the two sample sets $\mathcal{I}$ and $\mathcal{A}$ as

---

[4]In these works the regularization terms were motivated by maintaining input sensitivity and bounding the expected loss increase, respectively.

[5]We present a preposition showing that the interval incorporating the gradient norm decreases to the norm of $\mu$ with increasing amount of samples in supplement A.

$$\mathbb{E}_{\mathcal{I},\mathcal{A}}\left[\|\nabla_x f^{\mathcal{I}}_{y-c}(x)\|_2 \cdot \cos(\alpha_c^{\mathcal{I},\mathcal{A}})\right] = \mathbb{E}_{\mathcal{I},\mathcal{A}}\left[\|\nabla_x f^{\mathcal{I}}_{y-c}(x)\|_2 \cdot \frac{\langle -\nabla_x f^{\mathcal{I}}_{y-c}(x), \delta^{\mathcal{A}}\rangle}{\|\nabla_x f^{\mathcal{I}}_{y-c}(x)\|_2 \cdot \|\delta^{\mathcal{A}}\|_2}\right]$$

$$= \mathbb{E}_{\mathcal{I},\mathcal{A}}\left[\langle -\nabla_x f^{\mathcal{I}}_{y-c}(x), \frac{\delta^{\mathcal{A}}}{\|\delta^{\mathcal{A}}\|_2}\rangle\right] = \sum_{i=1}^{p}\mathbb{E}_{\mathcal{I}}\left[-\nabla_{x_i} f^{\mathcal{I}}_{y-c}(x)\right]\cdot \mathbb{E}_{\mathcal{A}}\left[\frac{\delta_i^{\mathcal{A}}}{\|\delta^{\mathcal{A}}\|_2}\right] \,,$$

where the last equation holds due to the independence of $\delta^{\mathcal{A}}$ and $-\nabla_x f^{\mathcal{I}}_{y-c}(x)$. The expectation of $\frac{\delta^{\mathcal{A}}}{\|\delta^{\mathcal{A}}\|_2}$ does not depend on the samples taken during inference and the expectation of the estimate of the derivative w.r.t. the $i$-th input $\mathbb{E}_{\mathcal{I}}\left[-\nabla_{x_i} f^{\mathcal{I}}_{y-c}(x)\right]$ is the same for different amounts of samples. Therefore, the expectation of the denominator does not change when changing the sample size during inference. This observation explains why the robustness of a stochastic classifier does not depend on the amount of samples taken during inference and gives an justification for picking an arbitrary sample size that allows for a good trade-off between efficiency and reduction of the variance of the MC estimate used for prediction.

## 5 Experimental robustness analysis

In this section we empirically demonstrate that the findings of our theoretical analysis are transferable to SNNs and help to explain the mechanisms leveraging the experimentally observed robustness of previously proposed SNNs.

**Experimental setup**   Our experiments are conducted on two different image datasets: FashionM-NIST [Xiao et al., 2017] and CIFAR10 [Krizhevsky et al.].[6] For experiments on FashionMNIST we used feedforward neural networks (FNN) with two stochastic hidden layers, each with 128 neurons. We trained the FNN as a Variational Matrix Gaussian (VMG, the BNN proposed by Louizos and Welling [2016]) via variational inference or as an infinite mixture (IM) with the maximum likelihood objective proposed by Däubener and Fischer [2020], with matrix variate normal distribution placed over the weights. We also trained FNNs of the same architecture, where we added Gaussian noise with $\sigma^2 = \{0.05, 0.1\}$ to the input, by minimizing the cross-entropy. We refer to these models as *stochastic input networks* (SINs) 0.05 and SIN 0.1, respectively. Note, that these networks correspond to the basic networks proposed for randomized smoothing by Lécuyer et al. [2019]. For experiments on CIFAR10 we trained two wide residual networks (ResNet) with MC dropout layers [Gal and Ghahramani, 2016] applied after the convolution blocks and dropout probabilities $p = 0.3$ and $p = 0.6$. If not specified otherwise we used 100 samples of $p(\Theta)$ for inference on all datasets and calculated adversarial attacks with the fast gradient (sign) method (FGM) [Goodfellow et al., 2015],

the cross-entropy loss for IMs, BNNs, and ResNets, the margin loss $\mathcal{L}_{\text{margin}}$ specified in section 3 for SINs, and the $\ell_2$-norm constraint based on the CleverHans repository [Papernot et al., 2018]. All experiments were run on a single NVIDIA GeForce RTX 2080 Ti. We refer the reader to supplement B for more details on the datasets, models, and the training procedure.

**Accuracy of robustness conditions**   We first investigated the practical transferability of the derived theorems. For enforcing the smoothness condition used in theorem 4.2, we built on the result from Yang et al. [2022], who showed that the $L$-smoothness parameter of a classifier $g$ : $x \rightarrow \mathbb{E}_\epsilon[f(x + \epsilon)]$ smoothed with random noise $\epsilon \sim \mathcal{N}(0, \sigma^2)$, is bounded by $L \leq 2/\sigma^2$. We therefore applied randomized smoothing during training of the models. For the BNN, on which

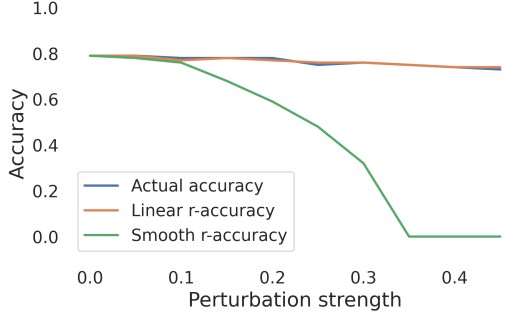

Figure 3: Adversarial accuracy of the smoothed BNN model for attacks based on 10 samples vs percentage of images with $\min_c r_c^{\mathcal{I}} > \|\delta^{\mathcal{A}}\|_2$ (smooth) and $\min_c \tilde{r}_c^{\mathcal{I}} > \|\delta^{\mathcal{A}}\|_2$ (linear) for the first 100 images from the FashionMNIST dataset.

---

[6]Additional experiments for CIFAR100 are presented in the supplement C.5.

we focus in this section due to space restrictions (results for the other networks look qualitatively similar and can be found in supplement C), we replaced each image in the batch by two noisy copies with Gaussian noise $\epsilon \sim \mathcal{N}(0, 0.1)$ which ensures $L \leq 20$. During prediction we estimated the expectation under Gaussian noise with 10 samples and also used 10 samples for calculating the FGM attack. We estimated the percentage of resulting attacks for which $\min_c r_c^{\mathcal{I}} > \|\delta^{\mathcal{A}}\|_2$ and $\min_c \tilde{r}_c^{\mathcal{I}} > \|\delta^{\mathcal{A}}\|_2$ and compared this to the adversarial accuracy (i.e. the percentage of perturbed samples classified correctly) in figure 3. The percentage of samples fulfilling the condition $\min_c r_c^{\mathcal{I}} > \|\delta^{\mathcal{A}}\|_2$ approaches zero with growing perturbation strength, indicating that the lower bound provided by $r_c^{\mathcal{I}}$ is rather loose due to the high (upper bound of the) L-smoothness constant. On the other hand the percentage of samples for which (3) is fulfilled is a good approximation of the real accuracy for small attack length, which suggests that the discriminant functions are approximately linear in a small neighborhood of the input.

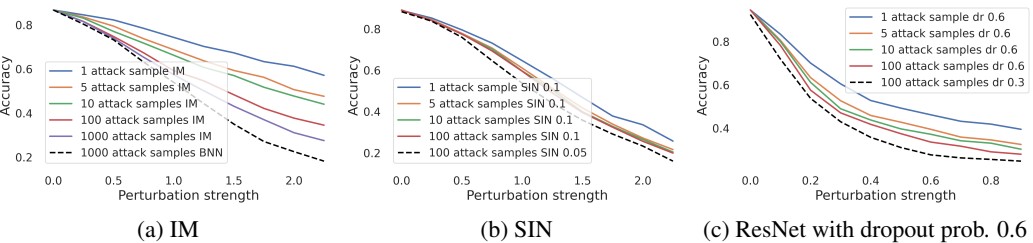

(a) IM  (b) SIN  (c) ResNet with dropout prob. 0.6

Figure 4: Accuracy under attack for different perturbation strength and amount of samples used for calculating the attack. The dashed line shows the adversarial accuracy for the models with the same architecture but less prediction variance.

**Stronger attacks by increased sample size**    In this section we investigate the effect of varying the amount of samples used for calculating the attack, e.g. for $S^{\mathcal{A}} \in \{1, 5, 10, 100, (1000)\}$. Figure 4 shows the resulting adversarial robustness of the IM, SIN 0.1, and the ResNet with dropout probability 0.6. The accuracy under attack decreases for all models with increasing amount of samples used for calculating the attack as conjectured. We found this to hold also for stronger attacks, i.e. attacks based on logits for IM and BNN (where we observe highly confident softmax predictions), attacks with $L_\infty$ constraint, or *projected gradient descent* (PGD) [Madry et al., 2018] attacks with 100 iterations as shown in supplement C. Note, that iterative attacks already increase the sample size due to their iterative nature, and therefore only few samples per iteration may sufficiently increase attack strength.

Simultaneously to the accuracy we estimated the corresponding values of $\cos(\alpha_j^{\mathcal{I},\mathcal{A}})$, for $j = \arg\min_c \tilde{r}_c^{\mathcal{I}}$, for the first 1,000 test images and depicted them in figure 5. It can be seen that the cosine values are increasing (which corresponds to decreasing angles) with growing sample size and that the larger the observed values, the lower the accuracy under attack as shown in figure 4. This observation is in accordance to our theoretical analysis which predicts that an increased amount of samples leads to higher cosine values and in turn to less adversarial robustness. Note however, that even when taking many samples $\cos(\alpha_j^{\mathcal{I},\mathcal{A}}) < 1$, which underlines the fact that the optimal attack direction can not be recovered due to the still existing stochasticity in the inference procedure. Further results on the angle under different amounts of samples during attack can be found in supplement C.2.

**Prediction variance as robustness indicator**    In this section we compare the properties of SNNs that have the same network architecture and a similar training procedure but different prediction variances. That is, we compare the BNN against the IM, SIN 0.05 against SIN 0.1, and the two ResNets with different dropout probabilities against each other. First, we estimated the standard deviation of the prediction and the average standard deviation of the gradient entries of the models by calculating the average of the empirical estimates over the first 1,000 examples from the respective test sets (see table 1). As expected, the IM, SIN 0.1, and ResNet with dropout probability 0.6 have a larger standard deviation of the prediction and hence prediction variance than their respective counterpart. The higher standard deviation of the prediction translates to an increased standard deviation of the gradient. This explains why we observe smaller cosine values for these models compared to their counterparts (see right most boxplots in figure 5 a) and c)) which in turn translate

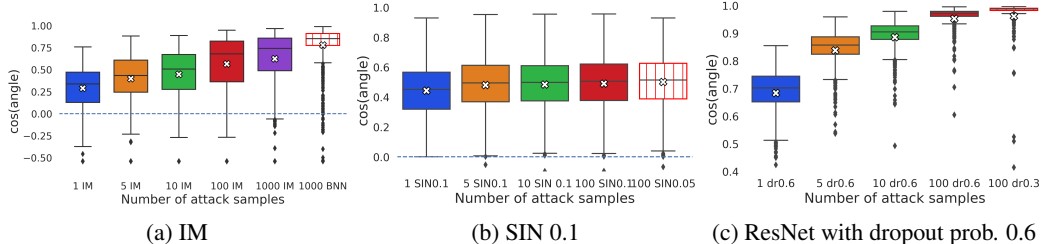

(a) IM          (b) SIN 0.1          (c) ResNet with dropout prob. 0.6

Figure 5: Cosine values for adversarial examples with perturbation strength $1.5$ for a) and b) and $0.3$ for c) and different amounts of samples. White crosses indicate mean values.

Table 1: Empirical standard deviation of the prediction (for the correct class) based on 1,000 single predictions (each with 1 inference sample) and average length and variance of the corresponding gradient. We report averages over the first 1,000 images from the respective test set.

|  | FASHIONMNIST | | | | CIFAR10 | |
|---|---|---|---|---|---|---|
|  | BNN | IM | SIN 0.05 | SIN 0.1 | DR 0.3 | DR 0.6 |
| AVG. STD OF PREDICTIONS | 0.0186 | 0.0473 | 85.4657 | 186.5599 | 0.0146 | 0.0196 |
| AVG. GRADIENT LENGTH | 0.5218 | 0.5044 | 49.0120 | 48.8312 | 0.0713 | 0.0599 |
| AVG. STD OF GRADIENT | 0.0316 | 0.0959 | 1.5741 | 1.8323 | 0.0000 | 0.0000 |

to an increased robustness (as indicated by the dotted accuracy curves in figure 4). We also found that the length of the mean of the gradient that we estimated based on 1,000 samples is smaller for the models with higher variance.[7] This might be another reason for the observed smaller angles as discussed in the previous section.

**Robustness in dependence of the amount of samples used during inference** In practice the amount of samples $S^{\mathcal{I}}$ drawn during inference is fixed to an arbitrary number. In this section we investigate the impact of varying $S^{\mathcal{I}} \in \{1, 5, 10, 100\}$ which are values frequently used. First, we observed that only few samples are necessary to get reliable predictions on clean data for all models as shown by the test accuracies in table 2. Second, we found that increasing the amount of samples did not affect the adversarial accuracy. This can be explained by the observation that the decrease of $\alpha_c^{\mathcal{I},\mathcal{A}}$ caused by increasing the sample set is counterbalanced by an simultaneous decrease of the average norm of the gradient estimate, as can be seen by inspecting the results in figure 6. That is, the product $\|\nabla_x f_{y-c}^{\mathcal{I}}\|_2 \cdot \cos(\alpha_c^{\mathcal{I},\mathcal{A}})$ stays approximately the same regardless of the inference sample size as predicted by our theoretical analysis. Interestingly, the effect of increasing the number of samples during inference has almost no effect on the prediction margin (more results are shown in supplement C.4).

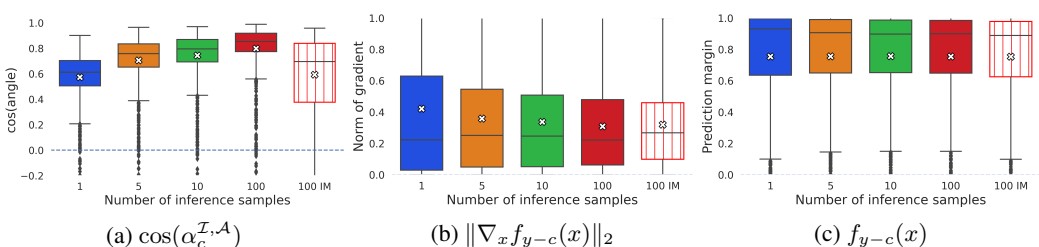

(a) $\cos(\alpha_c^{\mathcal{I},\mathcal{A}})$      (b) $\|\nabla_x f_{y-c}(x)\|_2$      (c) $f_{y-c}(x)$

Figure 6: Values of single factors of $\tilde{r}_c^{\mathcal{I}}$ in dependence of varying number of samples used during inference for the BNN trained on FashionMNIST.

---

[7]For SINs, it is known that a higher variance used during training relates to a lower Lipschitz-constant [e.g. Yang et al., 2022, Salman et al., 2019] which leads to a stronger smoothing effect.

Table 2: Test set and adversarial accuracy with 100 samples during the attack and allowed perturbation strength of $1.5$ on FashionMNIST and $0.3$ on CIFAR10 for increasing number of samples used during prediction. We estimated the average accuracy 10 times and report the average and standard deviation.

| $|\mathcal{I}|$ | FASHIONMNIST | | | | CIFAR10 | |
|---|---|---|---|---|---|---|
| | BNN | IM | SIN 0.05 | SIN 0.1 | DR 0.3 | DR 0.6 |
| | TEST SET ACCURACY | | | | | |
| 1 | $83.03 \pm 0.12$ | $79.15 \pm 0.27$ | $86.98 \pm 0.17$ | $85.76 \pm 0.17$ | $92.08 \pm 0.13$ | $92.68 \pm 0.15$ |
| 5 | $84.81 \pm 0.14$ | $84.29 \pm 0.21$ | $88.21 \pm 0.09$ | $87.49 \pm 0.15$ | $92.57 \pm 0.05$ | $93.51 \pm 0.09$ |
| 10 | $85.03 \pm 0.13$ | $85.02 \pm 0.18$ | $88.47 \pm 0.06$ | $87.92 \pm 0.10$ | $92.66 \pm 0.06$ | $93.60 \pm 0.06$ |
| 100 | $85.21 \pm 0.06$ | $85.72 \pm 0.08$ | $88.63 \pm 0.07$ | $88.27 \pm 0.06$ | $92.74 \pm 0.03$ | $93.69 \pm 0.04$ |
| | ADVERSARIAL ACCURACY | | | | | |
| 1 | $35.76 \pm 0.92$ | $44.67 \pm 1.19$ | $36.15 \pm 0.79$ | $40.63 \pm 0.79$ | $42.64 \pm 0.39$ | $47.18 \pm 0.70$ |
| 5 | $36.19 \pm 0.47$ | $47.08 \pm 0.73$ | $35.92 \pm 0.51$ | $40.57 \pm 0.44$ | $42.73 \pm 0.31$ | $47.00 \pm 0.38$ |
| 10 | $36.30 \pm 0.50$ | $47.23 \pm 0.62$ | $36.06 \pm 0.41$ | $39.93 \pm 0.66$ | $42.69 \pm 0.37$ | $47.01 \pm 0.33$ |
| 100 | $36.50 \pm 0.39$ | $47.96 \pm 0.24$ | $35.91 \pm 0.23$ | $39.67 \pm 0.29$ | $42.82 \pm 0.17$ | $47.02 \pm 0.20$ |

# 6 Conclusion

In this work we have stressed the fact that stochastic neural networks (SNNs) (and stochastic classifiers in general) often depend on samples and thus their predictions are random variables themselves. For gradient-based adversarial attacks this means that the attack is calculated based on one realization of the stochastic network (which depends on multiple samples of the random variables used in the network) and applied to another which is used for inference. We derived a sufficient condition for this inference network to be robust against the calculated attack. This allowed us to identify the factors that lead to an increased robustness of stochastic classifiers: i) larger prediction margins ii) a smaller norm of the gradient estimates and iii) higher angles between the attack direction and the direction to the closest decision boundary during inference. The observed angles depend inverse proportionally on the norm of the expected gradient and proportionally on the variance of the gradient estimates. This variance can be reduced by increasing the sample size. These insights enable us to explain previously reported empirical findings for SNNs from a geometrical perspective, e.g. that the robustness of SNNs is higher than the robustness of their deterministic counterparts even for strong attacks that are based on several samples [He et al., 2019, Eustratiadis et al., 2021], why regularization of the gradient variance [Bender et al., 2020], norm of the mean gradient, and angle [Lee et al., 2022] improves the adversarial robustness, and last but not least why increasing the sample size during attack is important to exploit its potential [Athalye et al., 2018]. Therefore, our work poses a general applicable and simple framework, that helps understanding the mode of operation of existing stochastic defense mechanisms, even if they were motivated from a different point of view. Moreover, we derived a justification for the common practice of choosing the sample size during inference in a way that balances its prediction certainty against the computational cost. Finally, we believe our findings will be useful to evaluate and compare the robustness of different models, since they point out that they might require different amounts of samples during attack to sufficiently reduce variance and hope that they will help to improve the robustness of stochastic classifiers in future.

**Potential negative societal impact and limitations.** Since we do not propose a new attack strategy, but contribute to a better understanding of the robustness of SNNs and advise to cautiousness when determining sample sizes, we do not see negative impact.

# Acknowledgments

We would like to thank Denis Lukovnikov for his useful comments on our work and beautifying our graphics. This work is funded by the Deutsche Forschungsgemeinschaft (DFG, German Research Foundation) under Germany's Excellence Strategy - EXC 2092 CASA - 390781972.

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
