# OpenReview forum: "How Sampling Impacts the Robustness of Stochastic Neural Networks"
_NeurIPS.cc/2022/Conference — NeurIPS 2022 Accept_

### Official Review · Reviewer_kUU3 · 2022-07-10

**Rating:** 7
**Confidence:** 4
**Soundness:** 3 good
**Presentation:** 4 excellent
**Contribution:** 3 good

**Summary:**

The authors perform a systematic study on the robustness of stochastic neural networks (SNNs). They make several theorectical corollaries: 1) In order to increase the probability of a successful attack against a SNN, the attacker need to increase the amount of samples to estimate the gradient. 2) The defender need to increase gradient variance and decrease the norm of the mean gradient, in order to improve the robustness of a SNN. 3) The robustness of a stochastic classifier does not depend on the amount of samples taken during inference.
The authors also conuduct extensive experiments on Fashion-MNIST and CAIFAR-10 datasets to emperically verify the correctness of these corollaries.

**Questions:**

The paper "Certified Adversarial Robustness via Randomized Smoothing" uses randomized smoothing to provide a robustness certificate for a neural network. A neural network with randomized smoothing can be seen as a type a stochastic neural network. How can Theorem 4.1 and Theorem 4.2 be used to certify robustness for a neural network with randomized smoothing? And how is it related to the robustness certificate in the paper "Certified Adversarial Robustness via Randomized Smoothing"?

**Limitations:**

They discuss the limitations and potential negative societal impact of their work in section 4 & 6.

**Strengths And Weaknesses:**

Strengths: The paper is written with good clarity and easy to follow. It also provide valuable theoretical insights on the robustness analysis of stochastic neural networks. The theoretical corollaries are well supported in the experiments.

Weaknesses: The datasets used in the experiments are relatively small.

---

> ### Author Response · Authors · 2022-08-02
> **Response to reviewer kUU3**
>
> Thank you very much for your review of our paper.
>
> **Weaknesses:** To incorporate more experiments, we will add the following results for the first 1,000 images of the CIFAR100 test set, where we used yet another way to derive a stochastic network: via Laplace approximation of the last layer to get an approximate last layer posterior.
> | Samples during attack  | Adversarial accuracy for $l_\infty$ constraint and perturbation strength 8/255 | Average $\cos ( \alpha^{\mathcal{I}, \mathcal{A}} ) \pm$ standard deviation |
> |---------|--------------|---------------|
> | 1 | 48.00      |0.2028 $\pm$ 0.112 |
> | 5 | 45.00 | 0.2550 $\pm$ 0.100 |
> | 10 | 43.90 | 0.2680 $\pm$ 0.095 |
>
>
> **Questions:** Regarding the relationship to “Certified Adversarial Robustness via Randomized Smoothing”, we kindly refer to the general response, as two other reviewers posed a similar question.

---

> > ### Comment · Reviewer_kUU3 · 2022-08-09
> > **Response to the rebuttal**
> >
> > I thank the authors for their response. My concerns are properly addressed and thus I will keep my rating of 7-Accept.

---

### Official Review · Reviewer_nbL1 · 2022-07-10

**Rating:** 3
**Confidence:** 4
**Soundness:** 2 fair
**Presentation:** 3 good
**Contribution:** 1 poor

**Summary:**

The authors study the adversarial attacks for stochastic classifiers. The attacker has the disadvantage of not knowing the exact direction of the attack since the hypothesis is a random variable and it would change randomly every time that it is queried. It has been observed that increasing the sample size would decrease the attack success rate, which goes against intuition on its face.

A sufficient condition for the robustness of the stochastic classifier against the calculated attack is proposed. Based on this condition, we can identify the factors that lead to an increased robustness of stochastic classifiers:
1- Larger expected prediction margins
2- A smaller expected norm of the gradient estimates
3- Higher angles between the attack direction and the direction to the closest decision boundary during inference.

The observed angles depend inverse proportionally on the norm of the expected gradient and proportionally on the variance of the gradient estimates. This variance can be reduced by increasing the sample size. The authors proposal explains the previously reported empirical findings for SNNs from a geometrical perspective, that the robustness of SNNs is higher than the robustness of their deterministic counterparts even for strong attacks that are based on several samples, why regularization of the gradient variance, norm of the mean gradient, and angle improves the adversarial robustness, and why increasing the sample size during attack is important to exploit its potential.

**Questions:**

What is the difference between a robust stochastic model and an stochastic model that have not converged in its derivatives?

**Limitations:**

The authors make a good effort to describe the limitations. However, they have not included the learning theory aspects of the phenomenon. The proposal is also positioned as an explanation for observations, where as it is more likely that the observations are not causally related to the phenomenon.

**Strengths And Weaknesses:**

- Originality:
The paper tries to bring insight on the phenomenon from the perspective of stochastic classifiers. The proposal is positioned as an explanation for the already observed aspects of the phenomenon is SNNs. From the three proposed conditions of robustness, large margin and the mean gradient norm are already studied in the literature. The angle is unique to the stochastic setting, and is the main result of the paper.

- Quality:
The authors have made the claims rigorous. The theorems and the results are intuitive enough. I have not read the supplementary material and the proofs. Nevertheless, the proposal lacks an analysis of the convergence of the classifiers in derivatives. The observations could be trivially explained if we assume that the derivatives of the trained SNNs have not converged in probability, or that the convergence is pointwise.

- Clarity:
The intuition behind the proposal is clear.

- Significance:
This is the my main concern with the paper. The main object of study in the paper is only definable for stochastic models. Consequently, it cannot be a good foundation for more general settings. It would be of help if the authors describe the proposal from the perspective of convergence in learning theory.

---

> ### Author Response · Authors · 2022-08-02
> **Response to reviewer nbL1**
>
> Thank you very much for the provided feedback.
>
> **Quality:** We are curious to understand your point of view that the observations are trivially explained if the derivatives have not converged in probability or if the convergence is pointwise. Could you further elaborate on this? Moreover, “not converged in probability or the convergence is pointwise” confuses us, since pointwise convergence implies convergence in probability.
>
> To better answer your question it would be helpful if you could share your definition/understanding of “convergence of derivatives”. Do you mean the property that the derivatives with respect to the parameters converge to zero during training for all training points? If so, how is this related to robustness?
> Note, that convergence of the gradients to zero on the training set would not result in zero gradients for test samples, and in the stochastic setting the mean gradient being zero would not even mean that the gradient for different parameter draws is zero.
>
> In general, our theory does not require convergence of the model during training. It applies for any given trained stochastic network whose prediction is based on MC samples. All we need in our theorems is that a data point $x$ (used to construct the adversarial example) is classified correctly (which can be assumed for a reasonable test accuracy of the network). Moreover, note that all models in our paper were trained for many epochs and the test set accuracies we archive are high and match with commonly used models in practice.
>
>
> **Significance:** In our work we indeed focus on stochastic classifiers, more specifically, on stochastic neural networks that rely on MC estimates for making their prediction. This is a very broad class of models by itself and therefore, we would argue that the analysis already covers a large number of models and is thus quite general. What kind of more general settings do you have in mind?

---

> > ### Comment · Reviewer_nbL1 · 2022-08-03
> > **elaboration on the review**
> >
> > - In general, our theory does not require convergence of the model during training.
> >
> > -- The main difficulty with your line of reasoning is that differentiable functions are not closed under uniform convergence [A]. Uniform convergence is a much stronger mode of convergence than pointwise convergence [B], which is in turn is stronger than convergence in probability [C]. Consequently, if we start from a random differentiable ANN and train the ANN, we are not guaranteed to end up with a differentiable function in the end, even when convergence is uniform. From this perspective even if we give the analysis benefit of the doubt of uniform convergence, it is still lacking in the assumption of differentiability.
> >
> > The mode of convergence is also related to integrability, a sequence of integrable functions that are converging pointwise are not guaranteed to be integrable [B]. In other words, Most of the logic in the proposal does not apply to the models that the authors study, because it is assumed that a differentiable and integrable ANN would still be differentiable and integrable after training. We recommend that the authors at the very least note that they are assuming uniform convergence in both value and the derivatives of the trained network. This is a very strong condition and might even be in opposition with the assumption that the trained network is weak to adversarial attacks. I don't think that any amount of empirical evidence can compensate for this flaw.
> >
> > - We are curious to understand your point of view that the observations are trivially explained.
> >
> > The explanation goes like this. Suppose that a trained network have not converged in its derivatives. Then, computing the empirical mean of samples of derivatives is analogues to computing the empirical mean of a Cauchy random variable. The divergence of the empirical mean as we increase the sample size is the first logical conclusion.
> >
> > -  “not converged in probability or the convergence is pointwise” confuses us.
> >
> > Sorry for the confusion. The correct sentence is “not converged, or the convergence is pointwise”.
> >
> > - Significance
> >
> > I agree that SNNs is a big family. But if the conclusions are not positioned from the perspective of convergence and learning theory, I cannot see how the arguments could be generalized to any deterministic hypothesis class. My current understanding of your proposal is that a degenerate (point mass) distribution over a class of single layer MLPs with 300 neurons is already out of the scope of the proposal.
> >
> > [A] https://en.wikipedia.org/wiki/Uniform_convergence#To_differentiability
> >
> > [B] https://people.math.wisc.edu/~angenent/521.2017s/UniformConvergence.html
> >
> > [C] https://en.wikipedia.org/wiki/Convergence_of_random_variables#Almost_sure_convergence

---

> > > ### Author Response · Authors · 2022-08-04
> > > **Response to reviewer nbL1**
> > >
> > > Dear reviewer,
> > >
> > > thank you very much for your elaborations and explanations. They helped us to understand your point of view and we agree that analyzing the properties of the convergence limit not only in SNNs but in general neural networks is an interesting problem. However, our analysis focuses on SNNs and investigates models obtained after finite training time, the scenario most relevant for practice. We hope we can now clarify the open questions in the following.
> > >
> > > **Convergence assumptions:**
> > >
> > > * ”Consequently, if we start from a random differentiable ANN and train the ANN, we are not guaranteed to end up with a differentiable function in the end, even when convergence is uniform. ”
> > >
> > > * “Most of the logic in the proposal does not apply to the models that the authors study, because it is assumed that a differentiable and integrable ANN would still be differentiable and integrable after training.”
> > >
> > > We agree that even under uniform convergence of a sequence of models $f_n$ during  training to a limit model $f$, it is not guaranteed that $\lim f_n = f$  is still a differentiable and integrable function.
> > > However, we believe that this is not a problem in the setting we analyze, since we focus on SNNs obtained after a finite training time (e.g. $f_n$ with $n<\infty$). If a differentiable model architecture (not taking into account the non-differentiable points induced by the ReLU activation functions at their respective origins) is picked, randomly initialized with small parameters, and the model is trained for some finite number of steps of stochastic gradient descent (or a related optimization technique) until a reasonable validation accuracy is reached, the model parameters are changed only moderately (assuming a reasonable learning rate and preventing exploding gradients e.g. with gradient clipping). Therefore, we believe the model is guaranteed to not leave the differentiable and integrable model class.
> > > If you are aware of an example demonstrating that one may end up with a non-differentiable and integrable function when training a BNN, a MC dropout network, or a SIN in such a finite  training time setting, this would be very helpful to get a better understanding of the problem you have in mind.
> > >
> > > We will add to our manuscript that we assume the trained SNNs to be integrable and differentiable such that the expected prediction and the gradient are well defined.
> > >
> > > **Explanation of observations:**
> > >
> > > * “Suppose that a trained network have not converged in its derivatives.”
> > >
> > > We still do not fully understand what “not converged in its derivatives” means here. Do you refer here to the derivatives of the model limit? Based on what you have explained before, do you mean that this finite limit model might be a random function for which an expectation of the derivatives does not exist (like in the case if it would follow a Cauchy distribution)?
> > > If the model obtained after training would have such a property this of course would be a problem, but as argued above we do not believe it is a realistic scenario in practice.
> > >
> > > * “The divergence of the empirical mean as we increase the sample size is the first logical conclusion”
> > >
> > > Please note that this is not what we observe in our experiments. We empirically see that with an increased sample size during the attack, the norm of the empirical mean gradient decreases (c.f. figure 10 in the supplement) and so does the accuracy under attack and the angle. Also the mean gradient norms (and the angles) seem to converge, which would not be the case for divergence of the mean estimate.
> > >
> > >
> > >
> > > **Significance:**
> > >
> > > Our analysis builds on the misalignment between attack direction and 'optimal attack direction in the inference network' that results from stochasticity. Therefore, generalizing the findings to the deterministic case does not make any sense to us.

---

> > > > ### Comment · Reviewer_nbL1 · 2022-08-04
> > > > **Examples in finite time settings**
> > > >
> > > > > investigates models obtained after finite training time, the scenario most relevant for practice.
> > > >
> > > > I cannot give you an example in the exact setting that you are considering, however Runge phenomenon [D] in interpolation is an example of how pointwise convergence would result in things going haywire in finite training sets (which is just another way of saying finite time for online learners). I believe that Runge-like phenomenon (which have been observed in many families of functions by the way, including the RBF family) is very relevant to practice in general, and the context of the present paper. Runge-like phenomenon have been observed in the context of classification [E] as well.
> > > >
> > > > > we focus on SNNs obtained after a finite training time.
> > > >
> > > > This reasoning is not enough; when exactly infinity starts? Where would you stop the training? The claims need to use stronger proofs. For example, to prove something about the hypothesis in the infinite time limit, first prove it for the finite case, and then use something like mathematical induction, or maybe a topological proof, to extend it to the infinite time limit. This way, you could argue that your theorem would be true after enough time/samples have passed/been observed.
> > > >
> > > > > We still do not fully understand what “not converged in its derivatives” means here.
> > > >
> > > > Take a look at the Weierstrass function [F] to get an image of how ill-defined derivatives can get when one is dealing with sequences of functions.
> > > >
> > > > > generalizing the findings to the deterministic case does not make any sense to us.
> > > >
> > > > Deterministic classifiers are just degenerate stochastic classifiers. In this degenerate case, the stochastic angle is meaningless, I get that. But we observe the phenomenon in the degenerate case. The effect of margin and the gradient norm are still present in the degenerate case. I skimmed the paper again, maybe I have missed it, but couldn't find how do you quantify the effect of stochastic angle compared to the effect of the gradient norm and the margin. That is the reason in my review I assert that the empirical results might not be causally related to the theorems.
> > > >
> > > > ---------------------------
> > > >
> > > > I think the best way forward would be for me to give a condition for when I would be convinced that the logic of the proposal is sound. If authors demonstrate that their logic applies to the Weierstrass function (add some random noise to make it stochastic if you prefer), and then come up with a sound process to quantify the effect of the proposed stochastic angle relative to the gradient norm and the margin, I will change my review to accept.
> > > >
> > > > [D] https://en.wikipedia.org/wiki/Runge%27s_phenomenon
> > > >
> > > > [E] https://arxiv.org/abs/2107.10599
> > > >
> > > > [F] https://en.wikipedia.org/wiki/Weierstrass_function

---

> > > > > ### Author Response · Authors · 2022-08-08
> > > > > **Reply to major concerns**
> > > > >
> > > > > Dear reviewer,
> > > > >
> > > > > thank you for the active discussion. To summarize, your main concerns are:
> > > > >
> > > > > 1) That our analysis of how incorporating stochasticity into neural networks impacts their robustness does not carry over to deterministic nets.
> > > > > 2) That there exist sequences of differentiable functions whose convergence limit is not a differentiable function, and assuming that the training of neural networks would result in such sequences our theory could not be applied to their limit (since the requirement of differentiability would not be fulfilled).
> > > > >
> > > > > We do not share
> > > > > - concern 1., since deterministic networks naturally do not contain any stochasticity. Therefore, analyzing their stochasticity is meaningless. As mentioned in our paper, in the deterministic setting $\tilde{r}^{\mathcal{I}}_c$ becomes equivalent to the shortest distance to the decision boundary.
> > > > > - concern 2., since, first of all, to the best of our knowledge there is no theoretical evidence that the sequences of differentiable neural networks encountered during training can have non-differentiable theoretical convergence limits. And even if one could construct such pathological cases, we do not think they are of any relevance in practice where one deals with finite sequences. We therefore argue that assuming differentiability of the stochastic networks is no hard restriction on the analyzed model class (and a property fulfilled by the networks usually encountered in practice) and that an analysis of the infinite update limit is (while being an interesting theoretical topic) out of scope for this paper.

---

> > > > > > ### Comment · Reviewer_nbL1 · 2022-08-09
> > > > > > **reply**
> > > > > >
> > > > > > > concern 1
> > > > > >
> > > > > > My concern is about the inverse of what you are considering. I am trying to say that stochasticity does not play any major role in the robustness of the stochastic classifier, and that the robustness of a stochastic classifier is completely described by the robustness of its mean, which is a deterministic classifier.
> > > > > >
> > > > > > > concern 2
> > > > > >
> > > > > > The critique is that the derivatives are ill-behaved, not that the derivatives does not exist.

---

> > > > > > > ### Author Response · Authors · 2022-08-09
> > > > > > > **Reply**
> > > > > > >
> > > > > > > Dear reviewer,
> > > > > > >
> > > > > > > we agree that analysing the robustness of the mean network, which indeed would be deterministic- would be as well highly interesting. However, this mean is usually not calculable in closed form and hence has to be approximated based on a computational reasonable amount of samples. The stochasticity induced by sampling increases the chance of an attack not being successful, because the attack most probably does not point into the ideal direction during inference. This effect is exactly what we investigate.
> > > > > > >
> > > > > > > Regarding the second, ill-behaved gradients are something people use to obfuscate the gradients to hinder (mostly unsuccessfully) adversarial attacks [1]. If you mean by “ill-behavior” that the derivatives of SNNs have a huge variance, this is exactly what makes attacking these networks difficult and which we capture by our theory, without claiming absolute robustness.
> > > > > > >
> > > > > > > [1] Athalye, A., Carlini, N., and Wagner, D. A., 2018, Obfuscated Gradients Give a False Sense of Security: Circumventing Defenses to Adversarial Examples, ICML

---

> > > > > ### Author Response · Authors · 2022-08-08
> > > > > **Reply to minor concerns**
> > > > >
> > > > > * We do not see a direct connection between the convergence to non-differentiable functions and the Runge phenomenon.
> > > > >
> > > > > * What do you mean by “when does infinity start”? Do you mean that for a sequence $f_{n}$ of differentiable functions there exists an $n'< \infty$ such that $f_{n'}$ is not differentiable? Do you have an example of such a sequence?
> > > > >
> > > > > * Our empirical findings perfectly fit what is expected from our theoretical analysis, and we are not aware of any other phenomenon that could cause these results. So what are the points that make you doubt the causality?
> > > > >
> > > > > * Regarding the Weierstrass function, we are aware that there are functions that are not differentiable- but trained neural networks -including BNNs and other SNNs- are (note that otherwise gradient-based training would also not be possible). Further, we do not see how the Weierstrass function can be used as a classifier such that the classifier is based a) on linear discriminant functions nor b) L-smooth discriminant functions. Therefore, our theory does not apply and hence your demand of proving otherwise is not fulfillable.

---

> > > > > > ### Comment · Reviewer_nbL1 · 2022-08-09
> > > > > > **reply**
> > > > > >
> > > > > > > We do not see a direct connection between the convergence to non-differentiable functions and the Runge phenomenon.
> > > > > >
> > > > > > The relation is that when the convergence of a sequence of differentiable functions to the pointwise limit is pointwise (which is a property of the learning rule), the derivatives of the function can become ill-behaved before (even much much before) reaching the pointwise limit. For example, look at this image [A] of a function that suffers from the Runge phenomenon. Also, look at figures 1 and 2 of [B] for examples in classification. All of these functions are differentiable in the sense that we can compute their derivative. The issue is that their derivative is ill-behaved. For example, I think that figure 1 of [B] has a counter example to result (iv) in the abstract of the paper:
> > > > > >
> > > > > > - (iv) why a higher gradient variance and shorter expected value of the gradient relates to a higher robustness.
> > > > > >
> > > > > > As we can see in the figure, the variance of the derivative of the trained Chebyshev classifier of degree 63 is very high and the symmetry in the graph suggests that the expected value of the derivative is zero.  Consequently, if this hypothesis is the mean of some stochastic classifier, and according to (iv), this stochastic classifier would be robust; which could not be farther from truth.
> > > > > >
> > > > > > > So what are the points that make you doubt the causality?
> > > > > >
> > > > > > I cannot decide if the results are caused by the non-robustness of the mean of the stochastic classifier (which is a deterministic classifier), or there is some genuine effect of stochasticity. It could be that you are measuring the properties of the non-robust mean classifier plus some random noise. For example in line 291-299, the paper describe why increasing the sample size does not change the adversarial accuracy and explains it using some complicated argument. Where as this observation could simply be explained if we assume that the mean classifier is not robust.
> > > > > >
> > > > > > In line 205, the paper asserts that it has explained why robustness of a stochastic classifier does not depend on the amount of samples taken, which I don't think that even needs an explanation to be honest, an alternative to the proposal in the paper is that the robustness of a stochastic classifier is mainly decided by the robustness of its mean, and that sampling is non-consequential when the sample size gets large enough.
> > > > > >
> > > > > > Table 2 in the paper actually is in line with the conclusion that no meaningful relation between sample size and robustness exists. For one, the change in the accuracy is insignificant. Second, while adversarial robustness increases with sample size for BNN and IM, other models in the experiment show a random behavior. Moreover, the changes in the mean accuracy of the models as we increase the sample size are smaller than the std of the estimation for most cases. I cannot see how anything meaningful could be inferred from this experiment.
> > > > > >
> > > > > > > Regarding the Weierstrass function, we are aware that there are functions that are not differentiable.
> > > > > >
> > > > > > Just to clarify again, I am not saying that there are some non-differentiable ANNs around. The derivatives of any finite ANN could always be computed without any troubles. The problem is that the derivatives might be ill-behaved, and you are using empirical means to reach some conclusions without compensating for the fact that derivatives could be ill-behaved. The Weierstrass function could act as a toy problem so that you could figure out how to make your argument work in the pathological cases. To make the function conform to your assumption, consider any $a$ and $b$ for which fractal behavior does not occur, then the function would be L-smooth. If the Weierstrass function is too hard, doing the same for the pathological examples in figures 1 and 2 of [B] is also enough. I would accept any other almost pathological example as well. But I would not accept the answer that this setting is not related to practice, because it is. The collection of functions that are differentiable at even a single point of interval $[0, 1]$ is a zero measure set. Given that ANNs are universal approximators, and hence dense in the set of square-integrable functions on $[0, 1]$, then we can conclude that if we sample a random ANN with enough neurons, almost surely that ANN is very pathological in its derivative. When I combine this observation with the fact that in practice we are considering something like $[0, 1]^{784}$ (e.g. Fashion-MNIST) and deep networks with 50 layers of neurons (ResNet 50), I cannot find your argument that these pathological functions are rare in practice compelling.
> > > > > >
> > > > > > [A] https://images.app.goo.gl/7QYsvfTLeMWYbNVY8
> > > > > > [B] https://arxiv.org/abs/2107.10599

---

> > > > > > > ### Author Response · Authors · 2022-08-09
> > > > > > > **Reply**
> > > > > > >
> > > > > > > We would like to use the example of Chebyshev maximum margin classifier to clarify the left misunderstandings: If we assume the Chebyshev classifier $g(x)$ to be the mean of a stochastic classifier $f(x|\theta)$, i.e. $\mathbb{E}(f(x|\theta)) = g(x)$, than the derivative of $g(x)$ is given by $\frac{d}{dx} g(x) = \frac{d}{dx}  \mathbb{E}(f(x|\theta))$, and this is for most $x$ unequal to zero. That is, because the variation of the derivative with respect to $x$ for different $x$ is not necessarily related to how it varies with regards to $\theta$.
> > > > > > >
> > > > > > > A case in which both derivatives (i.e. wrt. $x$ and $\theta$) are related is a SNN via randomised smoothing. But if one would take the Chebyshev classifier with degree 63 and add randomised smoothing to turn it into a stochastic classifier, the average prediction would also be smoothed and would look more  similar to the Bernstein maximum margin classifier (depending on the variance of the noise). Estimating the attack direction would however be more difficult due to the variance in the derivatives and decision boundaries, which would be perfectly described by our theory.
> > > > > > >
> > > > > > > Regarding the last open point: when talking about the universal approximation theorem you are again in an infinite limit (with respect to the number of hidden neurons). And as you said “The derivatives of any finite ANN could always be computed without any troubles.“ We fear that we have to agree to disagree on the impact of these theoretical considerations on our work.

---

### Official Review · Reviewer_izQZ · 2022-07-11

**Rating:** 6
**Confidence:** 3
**Soundness:** 3 good
**Presentation:** 3 good
**Contribution:** 3 good

**Summary:**

This paper drives a sufficient condition for the robustness of stochastic neural networks (SNNs). The theorems show that SNNs can classify adversarial examples correctly if the angle between adversarial perturbations and gradient of the inference estimate of SNNs satisfies specific properties. These theoretical results imply several strategies for attack and defense for SNNs.

**Questions:**

1. How is random smoothing related to the implications on SNN?

**Limitations:**

Yes

**Strengths And Weaknesses:**

Strength:
+ This paper gives a theoretical condition to show when SNNs can classify adversarial examples correctly.
+ Most parts of theorems and proofs are well-written and clear for understanding.
+ Empirical results are aligned with the theoretical results.

Weakness:

- Missing information in theorem 4.1: $c \neq y$ is used in proof but not in Theorem 4.1.  Supplementary material defines $\alpha$ in line 29, but Theorem 4.1 does not describe it.

- I think the theorems discussed in the paper are a bit too general and do not go deeper inside the design of SNNs from a probabilistic perspective:  The estimate used for attack and the estimate used for inference could be arbitrary two models with different decision boundaries.
It would be interesting to discuss more on how some properties (like the number of realizations) of SNNs influence the adversarial classification.

- A type of related work is not well discussed: Random smoothing is a popular certified defense method and could be also treated as a type of SNN. (Random variables only change the input and do not influence the weight). It would be good to mention and discuss how random smoothing fits into the proposed theory.

---

> ### Author Response · Authors · 2022-08-02
> **Response to reviewer izQZ**
>
> Thank you very much for your review and pointing us towards the missing information in the theorems which we will of course add.
>
> **Generality of theorems:** Indeed, the theorems are rather general and apply always when two different models for attack and inference are used. But unique to SNNs is that these theorems imply that even in a white-box attack scenario, where the attacker has full knowledge about the model parameters, the attack direction will not be optimal. The implications of this are analyzed in the second part of our theoretical section (and verified in our experiments), where we discuss and show how the amount of samples during attack and inference influence the robustness criteria. We would be happy to hear which further aspects you would be interested in.
>
> **Randomized smoothing:** Regarding the question about the relation of randomized smoothing to our findings we kindly refer to the general response as two other reviewers also asked about this connection.

---

> > ### Comment · Reviewer_izQZ · 2022-08-08
> > **Reply for the response**
> >
> > Thanks for the author's response and explanation.
> >
> > Based on the author's response and other reviewers' reviews, I decide to keep my score unchanged.

---

### Official Review · Reviewer_qGjN · 2022-07-11

**Rating:** 6
**Confidence:** 2
**Soundness:** 3 good
**Presentation:** 3 good
**Contribution:** 3 good

**Summary:**

This paper investigates the robustness of stochastic neural networks against adversarial attacks. To that end, it derives a sufficient condition on the robustness in terms of the decision margin, the gradient norms, and the angle between the decision boundary and the perturbation vector. The sufficient condition holds for linear and L-smooth classifiers and the paper provides some intuition as to why the conditions should generalize to arbitrary networks. The condition confirms various empirical approaches trying to improve robustness: more samples increase the attack performance, reducing the gradient variance and expected value increase the network’s robustness, and the number of samples used for inference is largely irrelevant.

**Questions:**

What are the implications for randomized smoothing methods aimed at increasing the certified robustness of neural networks?

**Limitations:**

The paper addresses the potential negative societal impact.

**Strengths And Weaknesses:**

The paper provides a novel theoretical perspective on the robustness of stochastic neural networks. It derives a sufficient condition on the robustness in terms of the decision margin, the gradient norms, and the angles between the perturbation vector and the decision boundary. As a consequence, the paper can analyze the influence of the different terms, which have been empirically used to increase robustness in prior work, on the robustness of neural networks. Thus, the paper presents a unified view of the robustness of stochastic neural networks, which can be useful for future research in this area.

Some typos:

- Line 97: the input *is* stochastic
- Line 269: theaccuracy -> the accuracy
- Line 271: many -> Many
- Caption of Table 1: correspondinggradient -> corresponding gradient

---

> ### Author Response · Authors · 2022-08-02
> **Response to reviewer qGjN**
>
> First of all, thank you very much for your review and pointing us typos in our paper which we of course will correct.
>
> Regarding your question about the implications of our findings for randomized smoothing methods we kindly refer to the general response as two other reviewers also asked about this connection.

---

### Author Response · Authors · 2022-08-02
**Explaining the relation to randomized smoothing**

Dear reviewers,

we thank you for your genuine feedback. Because the main questions in three of your reviews regard the relationship between our approach and randomized smoothing, we would like to use this general response to answer them altogether.

First, we would like to draw attention to the different ways of generating predictions for a data point $x$ in the randomized smoothing literature: Cohen et al. (2019) [1] use $\arg \max_{c \in \mathcal{Y}} P(f(x+ \epsilon) = c)$, where $\epsilon \sim \mathcal{N}(0, \sigma^2 I)$ and Lecuyer et al. (2019) [2] use $\arg \max_{c \in \mathcal{Y}} \mathbb{E}[A_c (x)]$, where $A$ is a randomized scoring function. In both approaches, a higher noise variance and a larger prediction margin lead to higher certified robustness bounds which is also valid in our findings. However, only the latter approach of calculating predictions is equivalent to the approach analyzed in our paper. Moreover, the networks we refer to as stochastic input networks (SIN) in our experiments are networks with Gaussian noise added to the input variables and hence equivalent to the simplest case of randomized smoothing proposed in [2] (we will make this more clear in our manuscript). Therefore, the theory of [2] can directly be applied to the SIN networks.

The certified robustness guarantees by randomized smoothing specify the radius of an $l_p$-ball in which the prediction does not change. Our results specify the robustness that results from the difficulty to identify the ideal attack direction and imply that even for perturbations outside this confidence ball, a gradient-based attack has a chance of not being successful. This chance is higher, the larger the variance of the gradients which is impacted by the noise used for randomized smoothing.

The general considerations of our paper can also be transferred to the case where predictions are based on the probability as in [1]. Since these probabilities are in practice approximated via sampling, the resulting decision boundaries are stochastic and hence differ during inference and attack. Unfortunately, as discussed in J3 in the supplement of [1] it is unclear how to effectively attack the probability based objective. An attack based on the expectation $\arg \max_{\delta: || \delta ||_2< \eta} E[\ell(f(x + \delta + \epsilon), c)]$ that results in a $\delta$ leading $x$ to cross a decision boundary w.r.t. the discriminant functions $f^{\mathcal{I}}$ does not necessarily result in the prediction of a different class if the prediction is based on the probability instead of the discriminant functions. For this reason, our theory can not directly be applied (but might be transferred to the probability based attack and decision boundaries if one finds an approach to estimate those).

-> In case of acceptance we will use the extra space to add a paragraph in the related work section to clarify the connection to randomized smoothing as done above.



[1] Cohen, J., Rosenfeld, E. & Kolter, Z.. (2019). Certified Adversarial Robustness via Randomized Smoothing. In: Proceedings of the 36th International Conference on Machine Learning

[2] Lécuyer, M., Atlidakis, V., Geambasu, R., Hsu, D. & Jana, S.. (2019) Certified Robustness to Adversarial Examples with Differential Privacy. In: IEEE Symposium on Security and Privacy

---

### Meta-Review · Area_Chair_f97e · 2022-08-27

**Recommendation:** Accept
**Confidence:** Less certain

**Metareview:**

All reviewers agree this paper studies an important problem and presents a principled analysis for robustness of SNNs. Empirical results, though limited, seems to be complementing the analysis well.  One reviewer rated the paper negatively. I find their concerns to be well addressed in the rebuttal phase. Overall this a borderline paper. I am suggesting acceptance and ecourage authors to update the draft to address all the reviewers comments.

**Award:**

No

---

### Decision · Program_Chairs · 2022-09-14

Accept